# Wavefront Aberration Sensor Based on a Multichannel Diffractive Optical Element

**DOI:** 10.3390/s20143850

**Published:** 2020-07-10

**Authors:** Svetlana N. Khonina, Sergey V. Karpeev, Alexey P. Porfirev

**Affiliations:** 1Image Processing Systems Institute—Branch of the Federal Scientific Research Centre “Crystallography and Photonics” of Russian Academy of Sciences, 443001 Samara, Russia; khonina@ipsiras.ru (S.N.K.); karp@ipsiras.ru (S.V.K.); 2Department of Technical Cybernetics, Samara National Research University, 443086 Samara, Russia; 3Department of Nanoengineering, Samara National Research University, 443086 Samara, Russia

**Keywords:** wavefront aberrations, Zernike polynomials, multi-channel diffractive sensor

## Abstract

We propose a new type of a wavefront aberration sensor, that is, a Zernike matched multichannel diffractive optical filter, which performs consistent filtering of phase distributions corresponding to Zernike polynomials. The sensitivity of the new sensor is theoretically estimated. Based on the theory, we develop recommendations for its application. Test wavefronts formed using a spatial light modulator are experimentally investigated. The applicability of the new sensor for the fine-tuning of a laser collimator is assessed.

## 1. Introduction

The problem of studying the deviation of a wavefront from the desired shape is one of the most significant in optics. There exist many well-known methods for solving this problem, and new techniques are being constantly developed. The most common and versatile among them is interferometry [1,2], which has unsurpassed accuracy and allows one to directly obtain a pattern of wavefront deviations at very large apertures. The accuracy of interferometers, especially heterodyne ones, exceeds λ/100. The disadvantages of interferometry are well known and include the complexity of decoding interferograms, the sensitivity of the measuring equipment to vibrations, and the need for the physical presence of a reference wavefront. At the initial stages of the development of optical production, the schlieren (or shadow) method was used to control spherical surfaces [3]; however, shadow patterns are difficult to quantify, and schlieren systems, like interferometers, must have high rigidity and be vibration-proof. The Hartmann method [4], which appeared later, differs from the previous techniques by the fact that the wavefront deviations are calculated from a set of subapertures, with some steps covering the full size of the region to be studied. Wavefront deviations are calculated using ray tracing data, with the rays passing through subapertures. A further development of the Hartmann method was a Shack–Hartmann wavefront sensor [5,6,7]. In this version of the sensor, the data on the wavefront deviations are transferred to the photodetector plane by installing a lens raster. Each lens forms a subaperture, for which an average wavefront deviation is calculated. Information about the wavefront phase within the subaperture is contained in the coordinates of the focused light spot. The main advantage of both the Hartmann method and the Shack–Hartmann wavefront sensor consists of the fact that there is no need to use a reference wavefront in calculations. However, these techniques are not exempt from disadvantages, such as sensitivity to vibrations and the initially discrete nature of the measurements. In this case, data on a part of the surface of the wavefront are inevitably lost. Many Shack–Hartman sensors currently being manufactured have no more than 10^3^ subapertures, which does not satisfy the requirements of many tasks. A new approach to the formation of microlens arrays based on mesoscale square cubic dielectric particles [8] was proposed, which will significantly increase the dimension of the lens raster.

One of the tools for describing wavefronts, in addition to the deviation patterns, is the decomposition of aberrations over various bases. The most famous decomposition bases include Zernike polynomials [9], as well as Seidel aberrations. Note that the generally accepted representation of wavefront aberrations [10,11,12], including in the individual optical system of the human eye, is a series of Zernike polynomials [7,13,14,15,16]. Aberrational representations are more efficient [17] in terms of data volumes and also allow one to make use of the wavefront features that are important for solving specific problems. The direct measurement of aberration coefficients is possible only for some types of aberrations. The calculation of the Zernike aberration coefficients [18,19,20,21] on the basis of a two-dimensional array of measured values of the wavefront deviations in each of the subapertures is provided in the data processing programs supplied with Shack–Hartmann sensors, as well as with ophthalmic aberrometers. However, it should be noted that due to the rather rough discretization of wavefront data, the calculation of high-order aberrations is difficult.

The active employment of Zernike polynomials for the representation of wave aberrations is stimulating the development of new sensors, including for the direct measurements of expansion coefficients by the Zernike basis. In this work, we propose a new sensor for measuring aberration coefficients based on a special multichannel diffractive optical element [22,23,24]. The developed sensor provides a sensitivity to wavefront deviations no worse than λ/20, is resistant to vibrations, and does not require the use of reference optical elements.

Diffractive optical elements for the integral calculation of the expansion coefficients of the amplitude–phase distributions of light fields over various bases [25,26,27,28], including the basis of Zernike functions [29,30,31], have been developed and used in fiber-optic sensors [32,33,34], for measuring the angular momentum of laser beams [35,36,37], for optical communication using mode and polarization (de)multiplexing [38,39,40,41,42,43,44], and in testing problems [45,46]. These elements make it possible to simultaneously obtain the values of the decomposition coefficients in the given elements of the photodetector matrix. In contrast to the Shack–Hartman sensor, in which the calculation of aberration coefficients requires the mathematical processing of a two-dimensional data array, the values of the aberration coefficients in multichannel diffractive optical sensors are proportional to the intensities of the diffraction maxima located at the photodetector matrix points with constant coordinates. Thus, the entire area of the tested beam is simultaneously involved in the formation of the values of each coefficient, while in Shack–Hartman sensors and especially in Hartman sensors, information about part of the wavefront area is not involved in the measurements and remains unknown. It should also be noted that the calibration function of the proposed sensor is substantially nonlinear, which leads to a decrease in the dynamic range of the aberrations being measured. However, this is quite enough for most practically significant cases of certification of optical systems; for example, it is believed that the average aberration should not exceed λ/10 for budget imaging systems and λ/100 for high-end systems.

## 2. Theoretical Background

The circular Zernike polynomials correspond to a complete set of orthogonal functions in polar coordinates (*r*, φ) in a circle of radius *r*_0_ [13]:(1)$$Z_{n}^{m}(r,\phi )=A_{n}R_{n}^{m}(r)\left\{\begin{array}{c} \cos(m\phi )\\ \sin(m\phi ) \end{array}\right\}$$
where $A_{n}=\sqrt{\left(n+1\right)/\pi }$ and $R_{n}^{m}\left(r\right)$ are the radial Zernike polynomials:(2)$$R_{n}^{m}(r)=\sum_{p=0}^{(n-m)/2}{\left(-1\right)}^{p}(n-p)!{\left[p!\left(\frac{n+m}{2}-p\right)!\left(\frac{n-m}{2}-p\right)!\right]}^{-1}{\left(\frac{r}{r_{0}}\right)}^{n-2p}$$

The expansion coefficients for the wavefront in the Zernike orthogonal functions (1) allow one to determine the deviations (aberrations) from the ideal wavefront [13,14,15,16,18,45,46].

Consider an aberrated wavefront in the form of a field:(3)g(r,ϕ)=exp[iψ(r,ϕ)]
whose phase is a superposition of Zernike functions:(4)$$\psi (r,\phi )=2\pi \alpha \sum_{n,m}b_{nm}Z_{n}^{m}(r,\phi )$$
where *b_nm_* are coefficients of the superposition.

The expansion coefficients of field (3) in basis (1) are calculated as follows:(5)$$c_{pq}=\int_{0}^{r_{0}}\int_{0}^{2\pi }g(r,\phi )Z_{p}^{q}(r,\phi )\;r\;dr\;d\phi$$

We represent field (3) as the following expansion:(6)$$\begin{array}{c} g(r,\phi )=\exp\left[i\psi (r,\phi )\right]=1+i\psi (r,\phi )-\frac{1}{2}\psi^{2}(r,\phi )-\frac{i}{6}\psi^{3}(r,\phi )+\ldots =\\ =1+i2\pi \alpha \sum_{n,m}b_{nm}Z_{n}^{m}(r,\phi )-2{\left(\pi \alpha \right)}^{2}{\left[\sum_{n,m}b_{nm}Z_{n}^{m}(r,\phi )\right]}^{2}+\ldots \end{array}$$

For small aberrations (i.e., the value of α), expression (6) can be significantly simplified:(7)$$g(r,\phi )\underset{\alpha \to 0}{\approx }1+i2\pi \alpha \sum_{n,m}b_{nm}Z_{n}^{m}(r,\phi )$$

Thus, if the aberrations are small enough to leave only the first two terms in expansion (6), then the field can be considered as a superposition of the functions themselves. In this case, the expansion coefficients of field (5) will be proportional to the coefficients in superposition (4):(8)$$\begin{array}{c} c_{pq}\underset{\alpha \to 0}{\approx }\int_{0}^{r_{0}}\int_{0}^{2\pi }\left(1+i2\pi \alpha \sum_{n,m}b_{nm}Z_{n}^{m}(r,\phi )\right)Z_{p}^{q}(r,\phi )\;r\;dr\;d\phi =\\ =A_{p}+i2\pi \alpha \cdot \delta_{pq,nm}\sum_{n,m}b_{nm}=A_{p}+i2\pi \alpha b_{pq}, \end{array}$$
where $A_{p}$ is the normalization constant, and $\delta_{pq,nm}$ is the Kronecker delta.

In this case, the type and magnitude of the aberrations can be detected using a multichannel filter matched with the Zernike functions [29,31,47]. Degtyarev et al. [47] showed that relation (7) can be used up to α = 0.3.

Physically, this corresponds to an average path difference between an ideal wavefront and a wavefront with an aberration coefficient of 0.3λ. For the visualization of the expansion coefficients, it is convenient to use a single index rather than a double one. Table 1 shows the correspondence of the Zernike functions (1) to a single index *l*.

It should be noted that the task of detecting small aberrations is topical, because the point scattering function (PSF) in this case slightly differs from the Airy pattern (diffraction spot) in the absence of aberrations (see Table 2).

Table 2 shows the phase distribution (wavefront) and PSF patterns in the presence of various aberrations corresponding to Zernike polynomials with different values of α. One can see that at α = 0.4, the intensity distributions in the focal plane of the lens (PSF) look approximately the same regardless of the aberration type and differ little from the diffraction spot. This fact makes it possible to establish the applicability criterion for a multichannel filter, matched with the Zernike functions, during the measurement process. The first column of Table 2 shows the distribution of the coefficients at α = 0.4; the correspondence of the index *l* to the decomposition coefficients (5) is shown in Table 1. It follows from Table 2 that only at α ≤ 0.4 (which corresponds to an average aberration coefficient of ≤0.4λ) can we detect (recognize) with confidence the aberration structure.

Approximations (7) and (8) become invalid with increasing α, and as the field expands in the Zernike basis, other coefficients will appear, except for those present in superposition (4).

Knowing this fact, we can determine to some extent how significant the level of aberration is. However, this can be done by measuring a sufficiently large number of factors. Given the need to do this using a single multi-channel diffractive sensor, it is desirable to optimize the number of necessary coefficients.

Note that the basis of the Zernike functions with trigonometric functions on angle (1) is not invariant to rotation, which is inconvenient in practical applications. Another representation of the Zernike functions is also well known:(9)$$Z_{n,m}(r,\phi )=B_{n}R_{n}^{m}(r)\exp\left(im\phi \right)$$
where $B_{n}=A_{n}$ at *m* = 0 and $B_{n}=A_{n}/2$ at *m* ≠ 0.

Obviously, functions (1) can be represented via a superposition of functions (9) and vice versa. Compared to (1), representation (9) is more convenient due to its in invariance to rotation. The Zernike basis in form (9) cannot be used in superposition (4); however, it can be conveniently used for the expansion of the optical field [31].

Basis (9) implies positive and negative values of the index *m*. The correspondence of a single index *l* to the pair indices (*n*, *m*) is shown in Table 3.

Below, we will consider some examples. For the convenience of further analysis, we write out explicit expressions for several Zernike polynomials (Table 4).

(1) Defocusing $Z_{2}^{0}(r,\phi )$
(10)$$g(r,\phi )=\exp\left[i2\pi \alpha Z_{2}^{0}(r,\phi )\right]=\exp\left[i2\pi \alpha A_{2}\left(2r^{2}-1\right)\right]$$

Taking into account expansion (6), field (10) can be represented as:(11)$$\begin{array}{c} \exp\left[i2\pi \alpha Z_{2}^{0}(r,\phi )\right]=1+i\pi \alpha Z_{2,0}(r,\phi )-{\left(\pi \alpha A_{2}\right)}^{2}\left(4r^{4}-4r^{2}+1\right)+\ldots =\\ =D_{0}+i\alpha D_{1}Z_{2,0}(r,\phi )-\alpha^{2}D_{2}Z_{4,0}(r,\phi )+\ldots \;\;\;, \end{array}$$
where $D_{j}$ are the reduced constants.

It can be seen from expression (11) that for large values of the parameter α, when the field (10) is expanded in the Zernike basis, in addition to coefficients with indices (0, 0) (*l* = 0) and (2, 0) (*l* = 4), there will appear a coefficient with the index (4, 0) (*I* = 12), as well as higher-order coefficients (see Figure 1).

Figure 1 shows the results from calculating the expansion coefficients of field (10) in basis (9) at different values of the parameter α. As can be seen, at small values of α, the coefficient (2, 0) (*l* = 4) is the largest (Figure 1a). With increasing α, the weight of the coefficient (4, 0) (*l* = 12), as well as that of the coefficient (6, 0) (*l* = 24), increases (Figure 1b). At a high level of defocusing, the weight of the coefficient (8, 0) (*l* = 40) is significantly enhanced (Figure 1c). Thus, the appearance of energy in high-order aberrations corresponds to a large level of available low-order aberration, and this effect can be detected by the optical expansion of the analyzed wavefront in the Zernike basis.

(2) Astigmatism $Z_{2}^{2}(r,\phi )$
(12)$$g(r,\phi )=\exp\left[i2\pi \alpha Z_{2}^{2}(r,\phi )\right]=\exp\left[i2\pi \alpha A_{2}r^{2}\cos2\phi \right]$$

Taking into account expansion (6), field (12) can be represented as:(13)$$\begin{array}{c} \exp\left[i2\pi \alpha Z_{2}^{2}(r,\phi )\right]=1+i\pi \alpha Z_{2,\pm 2}(r,\phi )-2{\left(\pi \alpha A_{2}\right)}^{2}r^{4}\left(1+\cos4\phi \right)+\ldots =\\ =D_{0}+i\alpha D_{1}Z_{2,\pm 2}(r,\phi )-\alpha^{2}\left[D_{2}Z_{2,0}(r,\phi )+D_{3}Z_{4,0}(r,\phi )+D_{4}Z_{4,\pm 4}(r,\phi )\right]+\ldots \;\;\;. \end{array}$$

The presence of $r^{4}\cos4\phi$ in (13) leads to the appearance of $Z_{4,\pm 4}(r,\phi )$, with $r^{4}$ resulting in the defocusing of various orders, in particular, $r^{4}=\left(R_{4}^{0}(r)+3R_{2}^{0}(r)+2\right)/6$.

Thus, expression (13) shows that for large values of the parameter α, in addition to coefficients with the indices (0, 0) (*l* = 0) and (2, ±2) (*l* = 3, 5), there will appear coefficients with the indices (2, 0) (*l* = 4), (4, 0) (*l* = 12), and (4, ±4) (*l* = 10, 14), as well as higher-order coefficients (see Figure 2).

As can be seen from the simulation, for small values of α, the coefficients (0, 0) (*l* = 0) and (2, ±2) (*l* = 3, 5) predicted in (13) are the largest (Figure 2a). With increasing α, the coefficients (2, 0) (*l* = 4) and (4, ±4) (*l* = 10, 14) become more significant (Figure 2b). It should be noted that with a high level of astigmatism, the field energy is distributed over a large number of coefficients, with defocusing (2, 0) (*l* = 4) being the most noticeable (Figure 2c).

(3) Coma $Z_{3}^{1}(r,\phi )$
(14)$$g(r,\phi )=\exp\left[i2\pi \alpha Z_{3}^{1}(r,\phi )\right]=\exp\left[i2\pi \alpha A_{3}\left(3r^{3}-2r\right)\cos\phi \right]$$

Field (14) can be represented as:(15)$$\begin{array}{c} \exp\left[i2\pi \alpha Z_{3}^{1}(r,\phi )\right]=\\ =1+i\pi \alpha Z_{3,\pm 1}(r,\phi )-{\left(\pi \alpha A_{3}\right)}^{2}\left(9r^{6}-6r^{4}+4r^{2}\right)\left(1+\cos2\phi \right)+\ldots =\\ =D_{0}+i\alpha D_{1}Z_{3,\pm 1}(r,\phi )-\\ -\;\alpha^{2}\left[D_{2}Z_{6,0}(r,\phi )+D_{3}Z_{6,\pm 2}(r,\phi )+D_{4}Z_{2,\pm 2}(r,\phi )+D_{5}Z_{4,\pm 2}(r,\phi )\right]+\ldots \;\;\;. \end{array}$$

Expression (15) is quite complex, but one can clearly see that aberrations with even *n =* 2, 4, 6 and *m* = ±2 additionally appear. Obviously, if we also take the cubic term into account, then additional odd aberrations with *m* = ±3 should appear.

As can be seen from the simulation, for small values of α, the coefficients (0, 0) (*l* = 0) and (3, ±1) (*l* = 7, 8) predicted in (15) are the largest (Figure 3a). With increasing α, the coefficients (6, 0) (*l* = 24), (6, ±2) (*l* = 23, 25), and (2, ±2) (*l* = 3, 5) also predicted in (15) become significant (Figure 3b). At a high coma level, the field energy is distributed over a large number of coefficients (Figure 3c). As expected, aberrations with higher angular multiplicity also appear in this case, namely (5, ±3) (*l* = 16, 19) and (7, ±3) (*l* = 30, 33). However, the coefficient (6, 0) (*l* = 24), corresponding to high-order defocusing, becomes the most noticeable (Figure 3c).

(4) Coma (trefoil) $Z_{3}^{3}(r,\phi )$
(16)$$g(r,\phi )=\exp\left[i2\pi \alpha Z_{3}^{3}(r,\phi )\right]=\exp\left[i2\pi \alpha A_{3}r^{3}\cos3\phi \right]$$

Field (16) can be represented as:(17)$$\begin{array}{c} \exp\left[i2\pi \alpha Z_{3}^{3}(r,\phi )\right]=\\ =1+i\pi \alpha Z_{3,\pm 3}(r,\phi )-{\left(\pi \alpha A_{3}\right)}^{2}r^{6}\left(1+\cos6\phi \right)+\ldots =\\ =D_{0}+i\alpha D_{1}Z_{3,\pm 3}(r,\phi )-\\ -\;\alpha^{2}\left[D_{2}Z_{2,0}(r,\phi )+D_{3}Z_{4,0}(r,\phi )+D_{4}Z_{6,0}(r,\phi )+D_{5}Z_{6,\pm 6}(r,\phi )\right]+\ldots \;\;\;. \end{array}$$

The presence of $r^{6}\cos6\phi$ in (17) leads to the appearance of $Z_{6,\pm 6}(r,\phi )$, i.e., a multiple increase in the angular dependence. The dependence $r^{6}$ can be described by the superposition $R_{6}^{0}(r)$, $R_{4}^{0}(r)$, $R_{2}^{0}(r)$ (i.e., defocusing of various orders). In addition, aberrations with the same angular dependence *m* = 3 but with a higher degree of radial polynomials (*n* > 3), may appear. The simulation confirms the theoretical analysis: for small values of α, the coefficients (0, 0) (*l* = 0) and (3, ±3) (*l* = 6, 9) are the largest (Figure 4a). With increasing α, the coefficients (5, ±3) (*l* = 16, 19) and (6, ±6) (*l* = 21, 27) increase, and the weight of the coefficients associated with defocusing (2, 0) (*l* = 4) also grows (Figure 3b). At a high level of aberrations, there also appear aberrations with a higher multiplicity (Figure 3c).

(5) Defocusing (quatrefoil) $Z_{4}^{4}(r,\phi )$
(18)$$g(r,\phi )=\exp\left[i2\pi \alpha Z_{4}^{4}(r,\phi )\right]=\exp\left[i2\pi \alpha A_{4}r^{4}\cos4\phi \right]$$

Field (16) can be represented as:(19)$$\begin{array}{c} \exp\left[i2\pi \alpha Z_{4}^{4}(r,\phi )\right]=1+i\pi \alpha Z_{4,\pm 4}(r,\phi )-\\ -{\left(\pi \alpha A_{4}\right)}^{2}r^{8}\left(1+\cos8\phi \right)+\ldots =D_{0}+i\alpha D_{1}Z_{4,\pm 4}(r,\phi )-\\ -\alpha^{2}[D_{2}Z_{2,0}(r,\phi )+D_{3}Z_{4,0}(r,\phi )+\\ +D_{4}Z_{6,0}(r,\phi )+D_{5}Z_{8,0}(r,\phi )+D_{6}Z_{8,\pm 8}(r,\phi )]+\ldots \;\;\;. \end{array}$$

Expression (19) is obtained similarly to the previous example.

The simulation results are shown in Figure 5: for small values of α, the coefficients (0, 0) (*l* = 0), (2, 0) (*l* = 4), and (4, ±4) (*l* = 10, 14) are the largest (Figure 5a). With increasing α, the coefficients (6, ±4) (*l* = 22, 26) and (8, ±8) (*l* = 36, 44) increase, and the weight of the coefficients (2, 0) (*l* = 4) associated with defocusing also grows (Figure 5b). At a large aberration level, aberrations with the same angular dependence but a higher degree of radial polynomials (6, ±4) (*l* = 22, 26) and (8, ±4) (*l* = 38, 42) become more significant and the effect of defocusing (2, 0) (*l* = 4), (4, 0) (*l* = 12), and (6, 0) (*l* = 24) is enhanced (Figure 5c).

The above examples allow us to identify the main trends associated with an increase in the level of aberrations. If the initial aberration has an angular dependence on the order of *m*, then aberrations appear with a multiple angular dependence of 2*m*, 3*m*. In addition, as a rule, there appears defocusing of various orders. The detection of defocusing is associated with an increase in the PSF area, which is always observed with an increase in the aberration level (see Table 2).

(6) Superposition $Z_{3}^{1}(r,\phi )+Z_{4}^{2}(r,\phi )$
(20)$$g(r,\phi )=\exp\left\{i2\pi \alpha \left[A_{3}\left(3r^{3}-2r\right)\cos\phi +A_{4}\left(4r^{4}-3r^{2}\right)\cos2\phi \right]\right\}$$

A theoretical analysis of expression (20) is rather difficult; therefore, we consider only the results of the numerical simulation shown in Figure 6.

For small values of α, the expected coefficients (0, 0) (*l* = 0), (3, ±1) (*l* = 7, 8), and (4, ±2) (*l* = 11, 13), as well as the additional defocusing (6, 0) (*l* = 24), are the largest (Figure 6a). With an increase in α, the weight of the coefficients (2, 0) (*l* = 4) and (6, 0) (*l* = 24) corresponding to defocusing increases (Figure 6b), which is explained by an increase in the PSF area.

The further enhancement of aberration leads to an almost uniform distribution of field energy over all coefficients, which should serve as a signal of a high level of wavefront distortion in measurements. In this case, other methods, including neural networks, need to be used to recognize and compensate for aberrations [48,49,50].

## 3. Experiments on Detection of Various Wavefront Aberrations Using a Zernike Filter

Figure 7a shows the optical scheme used in the experiment. The output from a solid-state laser (λ = 532 nm) was collimated using a system consisting of a pinhole (PH) with a hole diameter of 40 μm and a spherical lens (L_1_) (*f*_1_ = 250 mm). In this case, the lens (L_1_) was mounted on a linear translation stage and could be moved along the beam propagation axis with a step of 10 μm, which was done for subsequent experiments on measuring the defocusing of the collimated initial beam using our wavefront analyzers. Then, the expanded laser beam passed through a HOLOEYE LC 2012 transmissive spatial light modulator (SLM_1_) with a 1024 × 768 pixel resolution and a pixel size of 36 μm, which was used to form a wavefront with a required set of aberrations. Lenses (L_2_) (*f*_2_ = 150 mm), L_3_ (*f*_3_ = 150 mm), and a diaphragm (D) were used together to ensure the spatial filtering of the aberration-distorted beam formed by the first modulator. A HOLOEYE PLUTO VIS reflective spatial light modulator (SLM_2_) with a 1920 × 1080 pixel resolution and a pixel size of 8 μm was used to implement a phase mask of a multi-order analyzing diffractive optical element (DOE), which served to decompose the studied light field in terms of the Zernike polynomial basis. The laser beam reflected from the reflective light modulator using a beam splitter (BS)—a 4-f optical system of lenses L_4_ and L_5_ (*f*_4_ = 150 mm, *f*_5_ = 150 mm)—and a mirror (M) was directed to lens L_6_ (*f*_6_ = 350 mm), which focused it on the matrix of a ToupCam UCMOS08000KPB camera with a 3264 × 2448 pixel resolution and a pixel size of 1.67 μm. Part of the scheme—including the light modulator SLM_2_, lens L_6_, and video camera—is, in fact, a sensor. Other elements of the optical system are designed to simulate the studied beam and match the light modulator operating in a reflective regime with the rest of the system. The described optical system, in addition to checking the operability of the wavefront sensor, is used to calibrate the sensor by forming the studied beams with different aberrations and different α values using a controlled light modulator SLM_1_. In the industrial version of the sensor, the light modulator SLM_2_ will be replaced by a classical transmissive phase DOE made of a transparent material. This version of the sensor is easier to manufacture than the Shack–Hartman sensor and does not require vibration isolation. In the process of the measurements, the intensity patterns were stable both in the coordinates and in the measured intensity, although the optical system was not vibration-proof.

The phase mask of a 25-order analyzing DOE—which decomposes the incident light field in terms of the basis of Zernike functions (9) with numbers (*n*, *m*) = {(0, 0), (1, 1), (2, 0), (2, 2), (3, 1), (3, 3), (4, 0), (4, 2), (4, 4), (5, 1), (5, 3), (5, 5), (6, 0), (6, 2), (6, 4), (6, 6), (7, 1), (7, 3), (7, 5), (7, 7), (8, 0), (8, 2), (8, 4), (8, 6), (8, 8)}—is shown in Figure 7b. Although the complete basis (9) implies positive and negative values of the index *m*, taking into account a certain duplication of information in complex conjugate coefficients (with ±*m*), we used basis (9) only with positive *m* index values to reduce the number of diffraction orders (channels).

Figure 8 shows the experimental and numerical (using the Fourier transform) intensity distributions formed in the focal plane of lens L_6_ when the modulator SLM_2_ is illuminated by a laser beam with an aberrationless plane wavefront. The absence of any light peaks in the formed diffraction orders is clearly seen, which confirms the absence of aberrations in the illuminating beam.

The results of detecting various aberrations (including their combinations) using the developed multi-order analyzing diffraction sensors are demonstrated below. Figure 9 shows the experimentally obtained intensity distributions using a multichannel filter for various aberrations at α = 0.4. It can be seen that the theoretical analysis qualitatively agrees with the simulation. The intensities measured at the centers of diffraction orders can later serve as gauge parameters for studying wavefronts with previously unknown aberrations.

Figure 10 and Figure 11 show the experimentally obtained intensity distributions for various aberrations at α = 0.6 and 1, respectively. One can see the emergence of correlation peaks in diffraction orders corresponding to additional theoretically predicted aberrations. In all cases, the presence of defocusing (2, 0) is clearly visible.

It is convenient to evaluate the value of α in the wavefront under study and, accordingly, the applicability of the sensor according to the intensity distribution in the zero diffraction order corresponding to the PSF of the beam in question. Using the numerical simulation (see Table 2), we showed that for α ≤ 0.4, corresponding to the range of the sensor’s applicability, the PSF is close to the Airy pattern of a diffraction-limited system. With α > 0.4, the size of the PSF begins to increase due to the appearance of additional petals and rings, and the intensity in the center decreases (see Figure 10 and especially Figure 11). This serves as a criterion for exceeding the level of aberrations acceptable for the sensor.

Thus, we have experimentally confirmed that at α ≤ 0.4, the aberration structure can be confidently detected (recognized); however, with a further increase in α, recognition becomes problematic.

## 4. Experiments on the Collimator Fine-Tuning

One of the most important applications of the developed technique—the analysis of wavefront aberrations—is the testing and accurate adjustment of various optical components, for example, collimators. Figure 12 shows the intensity distributions obtained in the focal plane of lens L_6_ at various displacements of lens L_1_ forming the collimator along the beam propagation axis. One can see that when the lens is displaced from the initial plane *z* = 0, which corresponds to an ideal position of the lens (i.e., the position at which a laser beam with a plane wavefront is formed behind the lens) at the center of the diffraction order responsible for the Zernike polynomial with numbers (*n*, *m*) equal to (2, 0) (i.e., defocusing aberrations), an intensity peak appears. The intensity of this peak increases equally with the distance from the original plane *z* = 0 in both directions. Calculations show that for the used lens with a focal length of 250 mm and the illuminating beam with a diameter of about 5.4 mm, the deviation of the wavefront from the plane at the edge of the lens aperture at *z* = 0.5 mm is about λ/20. Thus, we can conclude that the sensitivity of the proposed sensor to local deviations of the wavefront is no worse than λ/20.

## 5. Conclusions

We studied, theoretically and experimentally, a new wavefront sensor based on a multi-channel element of computer optics. The established limit for measuring average wavefront aberrations is in the order of 0.4λ. For this range, the calibration function is independent of the values of the average wavefront aberration, and the calibration is performed once. In the process of adjusting the laser collimator, the wavefront deviation was recorded, which is maximum at the aperture edges and estimated to be no more than λ/20. The average aberration over the entire beam is much less. This sensitivity of the sensor allows one to detect defocusing of less than 0.5 mm for a laser radiation collimator with a relative aperture of about 1:50. Thus, in terms of sensitivity to aberrations, the proposed sensor is quite competitive with the Shack–Hartman sensor, but the proposed sensor does not require vibration isolation. The main advantage of the proposed sensor is that, unlike the Shack–Hartman sensor, where the aberration coefficients are calculated by mathematically processing a two-dimensional data array, the values of the coefficients in multichannel DOE sensors are directly proportional to the light intensities measured in individual pixels of the photodetector matrix with fixed coordinates. The entire area of the tested beam is simultaneously involved in the formation of the values of each coefficient, which reduces errors.

## Figures and Tables

**Figure 1 sensors-20-03850-f001:**
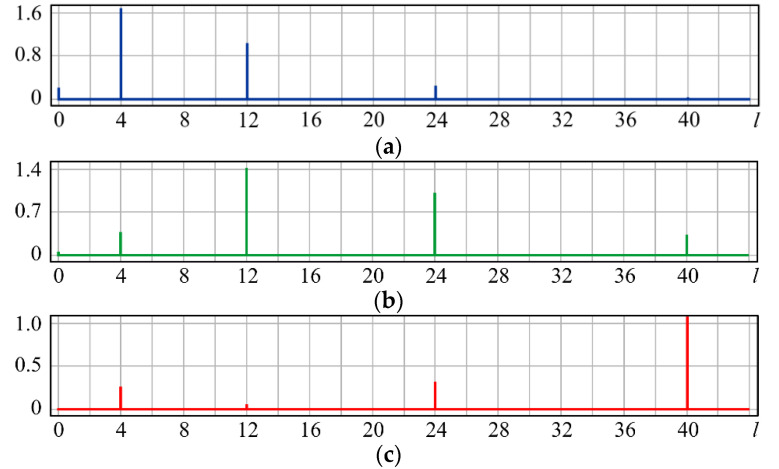
Wavefront expansion coefficients with defocusing $Z_{2}^{0}(r,\phi )$ of various levels: (**a**) α = 0.4, (**b**) α = 0.6, and (**c**) α = 1.

**Figure 2 sensors-20-03850-f002:**
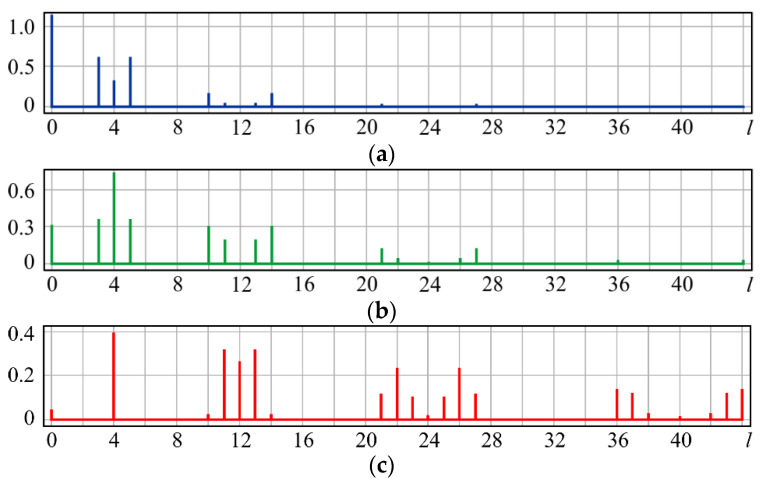
Wavefront expansion coefficients with astigmatism $Z_{2}^{2}(r,\phi )$ of various levels: (**a**) α = 0.4, (**b**) α = 0.6, and (**c**) α = 1.

**Figure 3 sensors-20-03850-f003:**
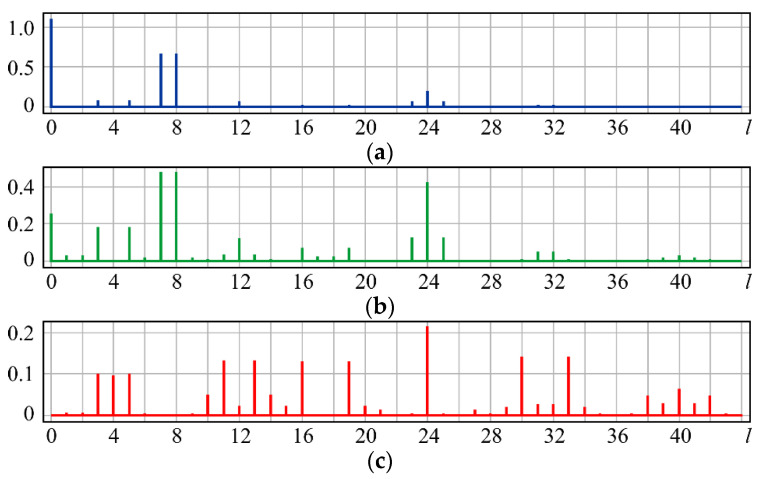
Wavefront expansion coefficients with coma $Z_{3}^{1}(r,\phi )$ of various levels: (**a**) α = 0.4, (**b**) α = 0.6, and (**c**) α = 1.

**Figure 4 sensors-20-03850-f004:**
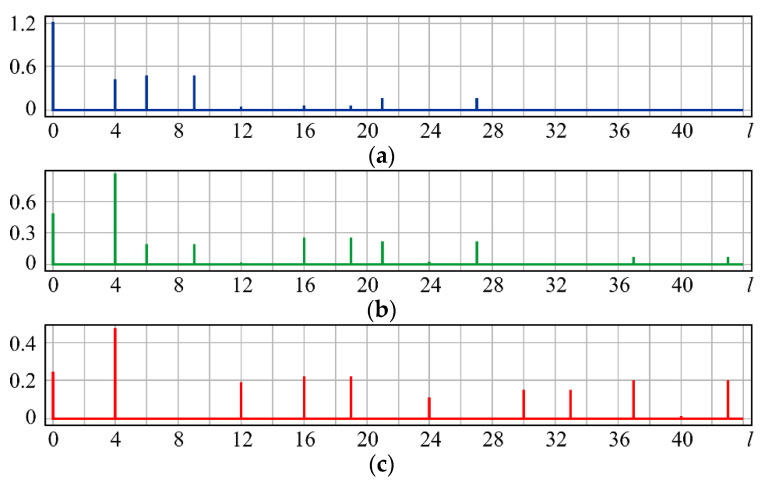
Wavefront expansion coefficients with coma (trefoil) $Z_{3}^{3}(r,\phi )$ of various levels: (**a**) α = 0.4, (**b**) α = 0.6, and (**c**) α = 1.

**Figure 5 sensors-20-03850-f005:**
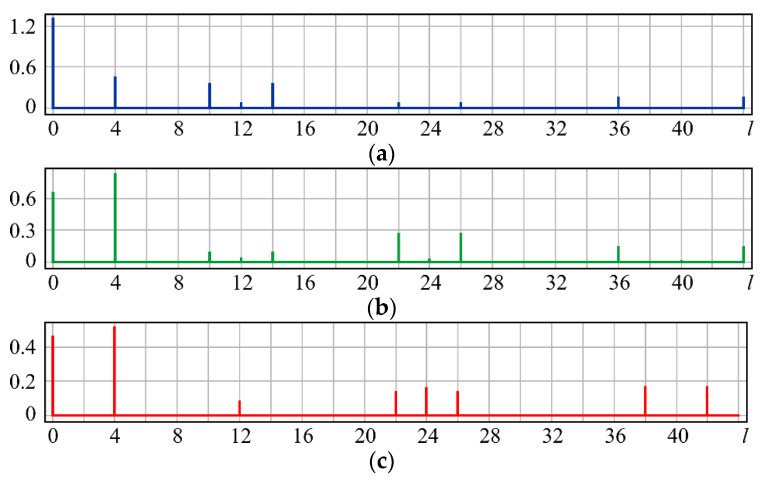
Wavefront expansion coefficients with coma (quatrefoil) $Z_{4}^{4}(r,\phi )$ of various levels: (**a**) α = 0.4, (**b**) α = 0.6, and (**c**) α = 1.

**Figure 6 sensors-20-03850-f006:**
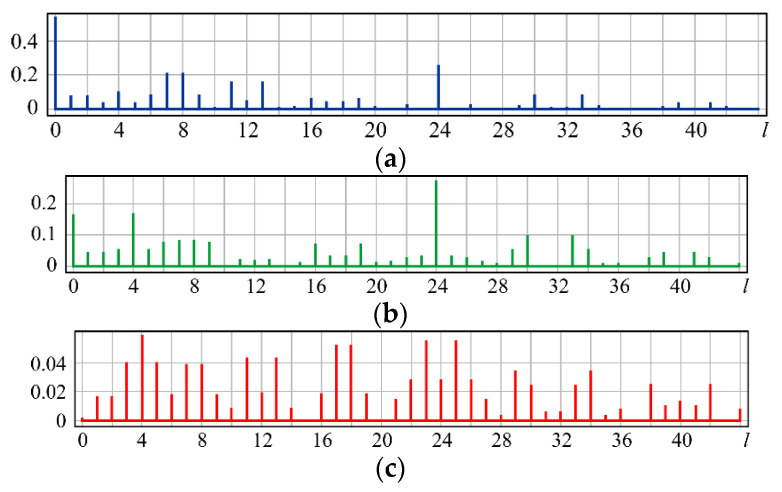
Wavefront expansion coefficients with aberration superposition $Z_{3}^{1}(r,\phi )+Z_{4}^{2}(r,\phi )$ of various levels: (**a**) α = 0.4, (**b**) α = 0.6, and (**c**) α = 1.

**Figure 7 sensors-20-03850-f007:**
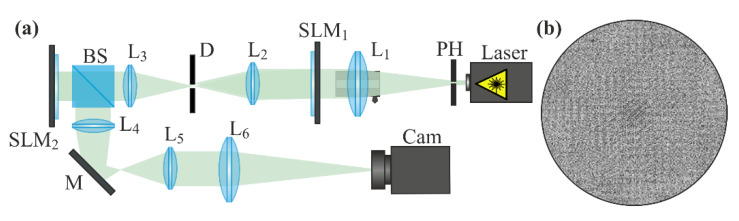
Detection of wavefront aberrations using a multi-channel analyzing diffractive optical element (DOE). (**a**) Schematic of the experimental setup: Laser is a solid-state laser (λ = 532 nm); PH is a pinhole (hole size of 40 μm); L_1_, L_2_, L_3_, L_4_, L_5_, and L_6_ are spherical lenses (*f*_1_ = 250 mm, *f*_2_ = 150 mm, *f*_3_ = 150 mm, *f*_4_ = 150 mm, *f*_5_ = 150 mm, and *f*_6_ = 350 mm); SLM_1_ is a transmissive spatial light modulator (HOLOEYE LC 2012); SLM_2_ is a reflective spatial light modulator (HOLOEYE PLUTO VIS); D is an aperture; BS is a beam splitter; M is a mirror; and Cam is a ToupCam UCMOS08000KPB video camera. (**b**) Phase mask of a 25-order analyzing DOE, which displays the incident light field in terms of the Zernike polynomials.

**Figure 8 sensors-20-03850-f008:**
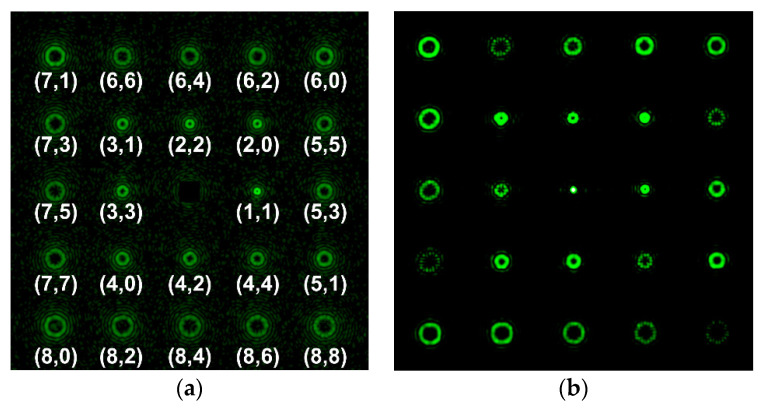
(**a**) Numerically and (**b**) experimentally obtained intensity distributions formed in the focal plane of lens L_6_ when SLM_2_ is illuminated by a laser beam with a plane wavefront. In the images of numerically obtained distributions, the zero diffraction order is cut out.

**Figure 9 sensors-20-03850-f009:**
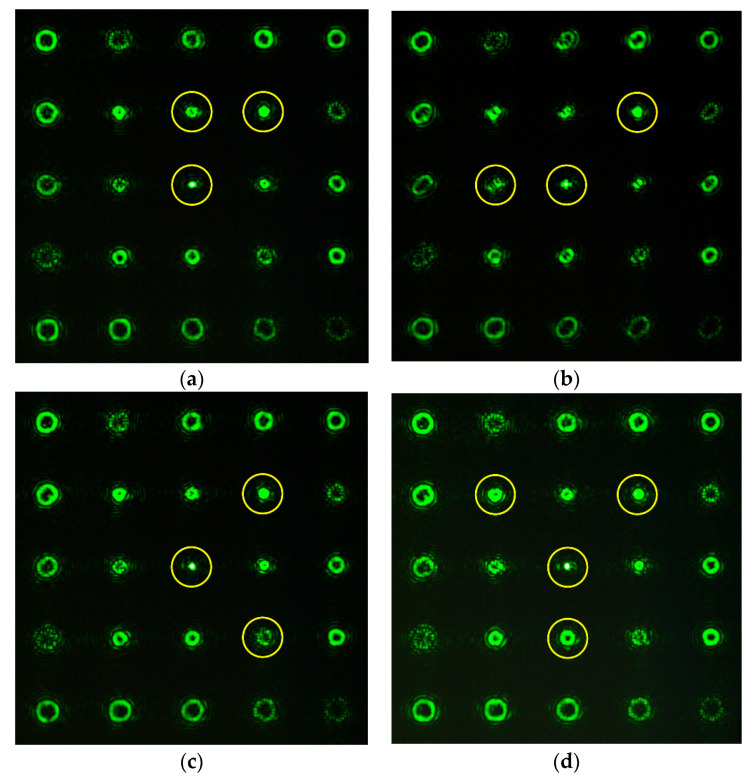
Experimentally obtained intensity distributions at α = 0.4 for (**a**) $Z_{2}^{2}(r,\phi )$ {(0, 0), (2, 0), (2, 2)}; (**b**) $Z_{3}^{3}(r,\phi )$ {(0, 0), (2, 0), (3, 3)}; (**c**) $Z_{4}^{4}(r,\phi )$ {(0, 0), (2, 0), (4, 4)}; and (**d**) $Z_{3}^{1}(r,\phi )+Z_{4}^{2}(r,\phi )$ {(0, 0), (2, 0), (3, 1), (4, 2)}.

**Figure 10 sensors-20-03850-f010:**
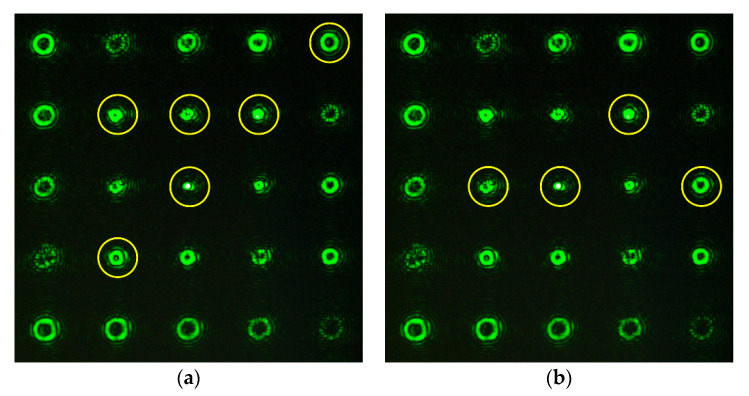
Experimentally obtained intensity distributions at α = 0.6 for (**a**) $Z_{3}^{1}(r,\phi )$ {(0, 0), (2, 0), (2, 2), (3, 1), (4, 0), (6, 0)}; and (**b**) $Z_{3}^{3}(r,\phi )$ {(0, 0), (2, 0), (3, 3), (5, 3)}.

**Figure 11 sensors-20-03850-f011:**
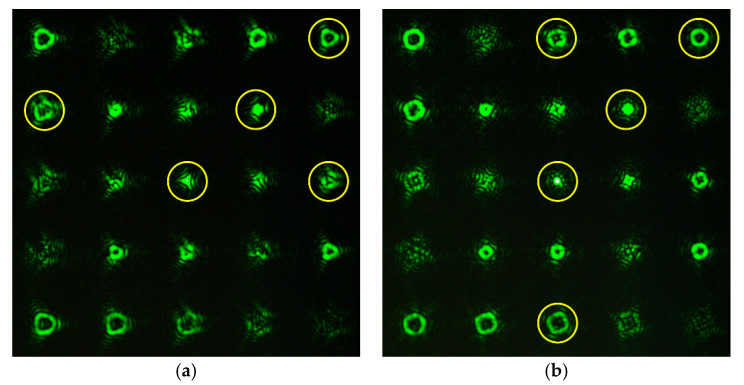
Experimentally obtained intensity distributions at α = 1 for (**a**) $Z_{3}^{3}(r,\phi )$ (0, 0), (2, 0), (5, 3), (6, 0), (7, 3); and (**b**) $Z_{4}^{4}(r,\phi )$ (0, 0), (2, 0), (6, 0), (6, 4), (8, 4).

**Figure 12 sensors-20-03850-f012:**
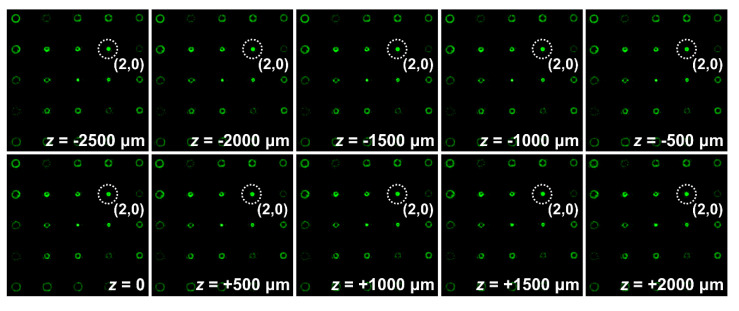
Intensity distributions obtained in the focal plane of lens L_6_ at different displacements of lens L_1_ forming the collimator. The dotted circles highlight the diffraction order corresponding to defocusing (Zernike polynomial $Z_{2}^{0}$).

**Table 1 sensors-20-03850-t001:** Correspondence of the Zernike functions (1) to a single index *l*.

*l*	0	1	2	3	4
$Z_{n}^{m}(r,\phi )$	$R_{0}^{0}(r)$	$R_{1}^{1}(r)\cos\phi$	$R_{1}^{1}(r)\sin\phi$	$R_{2}^{2}(r)\cos2\phi$	$R_{2}^{0}(r)$
*l*	5	6	7	8	9
$Z_{n}^{m}(r,\phi )$	$R_{2}^{2}(r)\sin2\phi$	$R_{3}^{3}(r)\cos3\phi$	$R_{3}^{1}(r)\cos\phi$	$R_{3}^{1}(r)\sin\phi$	$R_{3}^{3}(r)\sin3\phi$
*l*	10	11	12	13	14
$Z_{n}^{m}(r,\phi )$	$R_{4}^{4}(r)\cos4\phi$	$R_{4}^{2}(r)\cos2\phi$	$R_{4}^{0}(r)$	$R_{4}^{2}(r)\sin2\phi$	$R_{4}^{4}(r)\sin4\phi$

**Table 2 sensors-20-03850-t002:** The type of phase (wavefront) and point scattering function (PSF) in the presence of various aberrations corresponding to Zernike polynomials.

Expansion Coefficients	Input Phase and PSF (Intensity)		
	α = 0.4	α = 0.6	α = 1
$\psi (r,\phi )=0.8\pi R_{2}^{0}(r)$ 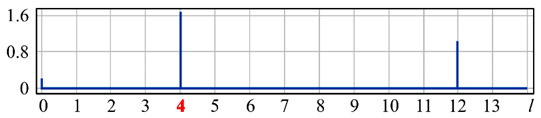	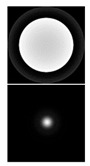	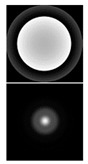	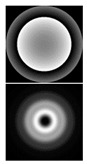
$\psi (r,\phi )=0.8\pi R_{2}^{2}(r)\cos2\phi$ 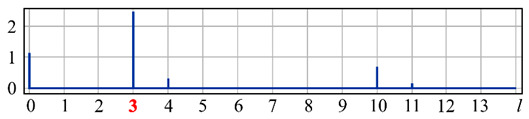	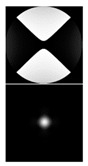	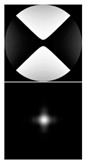	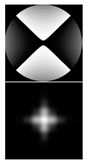
$\psi (r,\phi )=0.8\pi R_{3}^{1}(r)\cos\phi$ 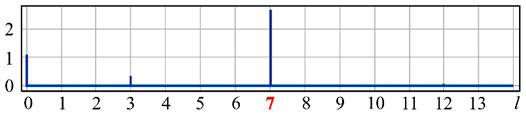	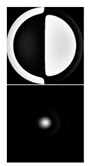	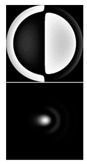	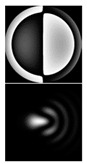
$\psi (r,\phi )=0.8\pi R_{3}^{3}(r)\cos3\phi$ 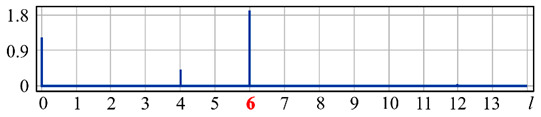	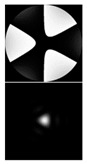	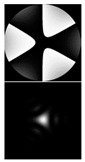	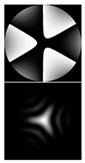
$\psi (r,\phi )=0.8\pi R_{4}^{0}(r)$ 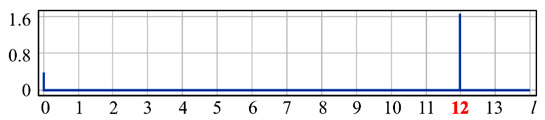	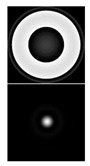	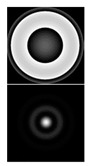	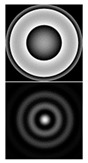
$\psi (r,\phi )=0.8\pi R_{4}^{2}(r)\cos2\phi$ 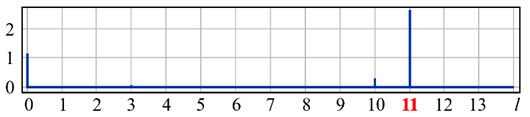	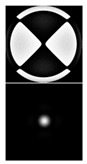	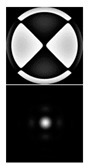	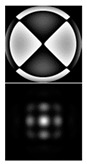
$\psi (r,\phi )=0.8\pi R_{4}^{4}(r)\cos4\phi$ 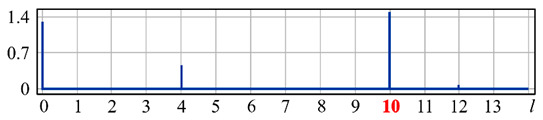	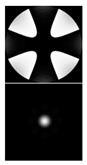	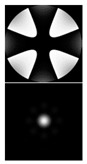	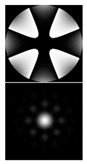
$\psi (r,\phi )=0.8\pi \left[R_{3}^{1}(r)\cos\phi +R_{4}^{2}(r)\cos2\phi \right]$ 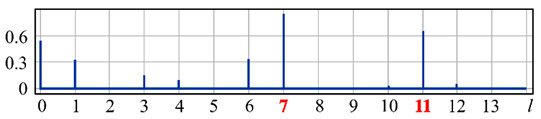	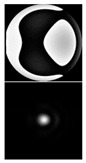	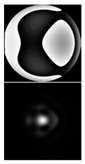	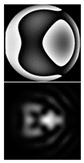

**Table 3 sensors-20-03850-t003:** Correspondence of the considered Zernike functions (9) to a single index *l*.

*l*	0	1	2	3	4	5	6	7	8
(*n*, *m*)	(0, 0)	(1, 1)	(1, −1)	(2, 2)	(2, 0)	(2, −2)	(3, 3)	(3, 1)	(3, −1)
*l*	9	10	11	12	13	14	15	16	17
(*n*, *m*)	(3, −3)	(4, 4)	(4, 2)	(4, 0)	(4, −2)	(4, −4)	(5, 5)	(5, 3)	(5, 1)
*l*	18	19	20	21	22	23	24	25	26
(*n*, *m*)	(5, −1)	(5, −3)	(5, −5)	(6, 6)	(6, 4)	(6, 2)	(6, 0)	(6, −2)	(6, −4)
*l*	27	28	29	30	31	32	33	34	35
(*n*, *m*)	(6, −6)	(7, 7)	(7, 5)	(7, 3)	(7, 1)	(7, −1)	(7, −3)	(7, −5)	(7, −7)
*l*	36	37	38	39	40	41	42	43	44
(*n*, *m*)	(8, 8)	(8, 6)	(8, 4)	(8, 2)	(8, 0)	(8, −2)	(8, −4)	(8, −6)	(8, −8)

**Table 4 sensors-20-03850-t004:** Explicit expressions for some Zernike polynomials.

(*n*, *m*)	$R_{n}^{m}(r)$	(*n*, *m*)	$R_{n}^{m}(r)$
(0, 0)	1	(4, 0)	$6r^{4}-6r^{2}+1$
(1, 1)	r	(4, 2)	$4r^{4}-3r^{2}$
(2, 0)	$2r^{2}-1$	(4, 4)	$r^{4}$
(2, 2)	$r^{2}$	(5, 1)	$10r^{5}-12r^{3}+3r$
(3, 1)	$3r^{3}-2r$	(5, 3)	$5r^{5}-4r^{3}$
(3, 3)	$r^{3}$	(5,5)	$r^{5}$

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
