# Peer review of "Wavefront Aberration Sensor Based on a Multichannel Diffractive Optical Element"

_sensors, 2020, doi:10.3390/s20143850_

Round 1

Reviewer 1 Report

The manuscript is well-written, it provides theory, simulations and experimental results. I suggest the manuscript for publication in its present form.

Author Response

We are thankful to reviewers for their useful comments and suggestions, which allow us to improve the quality of the manuscript making it more clear for readers. We believe the corrections made address the Reviewer’s concerns making the manuscript suitable for publication in the journal.

All changes in the manuscript are highlighted by yellow color.

 Reviewer 1

The manuscript is well-written, it provides theory, simulations and experimental results. I suggest the manuscript for publication in its present form.

Reply.

Thank you for your comment.

Reviewer 2 Report

Authors have studied a new wavefront sensor both theoretically and experimentally. The results in the manuscript are interesting and helpful to the researchers in this field. This review recommends an acceptance for a publication after minor revisions.

Line 42, change “no more that 103subapertures” to “no more than 103 subapertures”.

line 260, leave a space between “alpha” and “value”.

Author Response

We are thankful to reviewers for their useful comments and suggestions, which allow us to improve the quality of the manuscript making it more clear for readers. We believe the corrections made address the Reviewer’s concerns making the manuscript suitable for publication in the journal.

All changes in the manuscript are highlighted by yellow color.

 Reviewer 2

Authors have studied a new wavefront sensor both theoretically and experimentally. The results in the manuscript are interesting and helpful to the researchers in this field. This review recommends an acceptance for a publication after minor revisions.

Line 42, change “no more that 103subapertures” to “no more than 103 subapertures”.

line 260, leave a space between “alpha” and “value”.

Reply.

Thank you for your comments. We corrected it.

Reviewer 3 Report

In common the paper well written.

Some minor comments are:

In introduction it could be noted that in the newest Shack–Hartmann wavefront sensor raster based on mesoscale square cubic dielectric particles, the characteristic size of the sub-aperture is equal to the wavelength of the radiation being analyzed [DOI 10.1007/978-3-319-24253-8]. These subapertures form photonic jets array and are arranged in a staggered manner.

Sensor calibration procedure is described too briefly.

There are some misprints in the manuscript, please check.

Author Response

We are thankful to reviewers for their useful comments and suggestions, which allow us to improve the quality of the manuscript making it more clear for readers. We believe the corrections made address the Reviewer’s concerns making the manuscript suitable for publication in the journal.

All changes in the manuscript are highlighted by yellow color.

Reviewer 3

In common the paper well written.

Some minor comments are:

In introduction it could be noted that in the newest Shack–Hartmann wavefront sensor raster based on mesoscale square cubic dielectric particles, the characteristic size of the sub-aperture is equal to the wavelength of the radiation being analyzed [DOI 10.1007/978-3-319-24253-8]. These subapertures form photonic jets array and are arranged in a staggered manner.

Reply.

Thank you for your comment. We added the following text:

A new approach to the formation of microlens arrays based on mesoscale square cubic dielectric particles [8] was proposed, which will significantly increase the dimension of the lens raster.

Sensor calibration procedure is described too briefly.

Reply.

Thank you for your comment. In accordance with the described calibration procedure within the linear region of the sensor calibration function (average wavefront aberrations <0.4λ), calibration for different values of the average wavefront aberration is not required. We have added the following text:

For this range, the calibration function is independent on values of the average wavefront aberration and the calibration is performed once.

There are some misprints in the manuscript, please check.

Reply.

Thank you for your comment. We carefully checked the text again and corrected the misprints.

Reviewer 4 Report

This is an interesting paper that exploits the application of Zernike polynomials in diffraction theory. In the title, the term 'Zernike' appears more appropriate than 'Fourier'.

The authors should better distinguish their recapitulation of the literature from their own extensions/applications.

The experimental work and simulations are interesting enough.

It is a bit surprising that original papers doing more or less the same, with technology of 70 years ago, are not so well cited.

Zernike, F (1934) Beugungstheorie des Schneidenverfahrens und Seiner
Verbesserten Form, der Phasenkontrastmethode. Physica. 1/8: 689–704.

(also available reprinted in English)

Nijboer, B.R.A. The diffraction theory of optical aberrations. Part I: General discussion of the geometrical aberrations (1943) Physica, 10 (8), pp. 679-692.

Nijboer, B.R.A. The diffraction theory of optical aberrations. Part II: Diffraction pattern in the presence of small aberrations (1947) Physica, 13 (10), pp. 605-620. Nienhuis, K., Nijboer, B.R.A. The diffraction theory of optical aberrations. Part III: General formulae for small aberrations; experimental verification of the theoretical results (1949) Physica, 14 (9), pp. 590-604,IN1,605-608.

Evans, C J, Parks, R E, Sullivan, P J, Taylor, J S (1995) Visualization of surface figure by the use of Zernike polynomials. Applied Optics, 34/34:7815-7819.

Author Response

We are thankful to reviewers for their useful comments and suggestions, which allow us to improve the quality of the manuscript making it more clear for readers. We believe the corrections made address the Reviewer’s concerns making the manuscript suitable for publication in the journal.

All changes in the manuscript are highlighted by yellow color.

Reviewer 4

This is an interesting paper that exploits the application of Zernike polynomials in diffraction theory. In the title, the term 'Zernike' appears more appropriate than 'Fourier'.

Reply.

Thank you for your comment. We did it

The authors should better distinguish their recapitulation of the literature from their own extensions/applications.

The experimental work and simulations are interesting enough.

Reply.

Thank you for your comment. We modified the introduction and added the following text:

Active employment of Zernike polynomials for representation of wave aberrations stimulates the development of new sensors, including for direct measurements of expansion coefficients by the Zernike basis.

It is a bit surprising that original papers doing more or less the same, with technology of 70 years ago, are not so well cited.

Zernike, F (1934) Beugungstheorie des Schneidenverfahrens und Seiner Verbesserten Form, der Phasenkontrastmethode. Physica. 1/8: 689–704. (also available reprinted in English)

Nijboer, B.R.A. The diffraction theory of optical aberrations. Part I: General discussion of the geometrical aberrations (1943) Physica, 10 (8), pp. 679-692.

Nijboer, B.R.A. The diffraction theory of optical aberrations. Part II: Diffraction pattern in the presence of small aberrations (1947) Physica, 13 (10), pp. 605-620. Nienhuis, K., Nijboer, B.R.A. The diffraction theory of optical aberrations. Part III: General formulae for small aberrations; experimental verification of the theoretical results (1949) Physica, 14 (9), pp. 590-604,IN1,605-608.

Evans, C J, Parks, R E, Sullivan, P J, Taylor, J S (1995) Visualization of surface figure by the use of Zernike polynomials. Applied Optics, 34/34:7815-7819.

 Reply.

Thank you for your comment and useful references. We added them to the text.

Round 2

Reviewer 4 Report

The paper looks more complete now and can be published.